# Impact of Pituitary Autoimmunity and Genetic Disorders on Growth Hormone Deficiency in Children and Adults

**DOI:** 10.3390/ijms21041392

**Published:** 2020-02-19

**Authors:** Giuseppe Bellastella, Maria Ida Maiorino, Miriam Longo, Paolo Cirillo, Lorenzo Scappaticcio, Maria Teresa Vietri, Antonio Bellastella, Katherine Esposito, Annamaria De Bellis

**Affiliations:** 1Unit of Endocrinology and Metabolic Diseases, University of Campania “Luigi Vanvitelli”, 80138 Naples, Italy; gbellastella@hotmail.com (G.B.); mariaida.maiorino@unicampania.it (M.I.M.); miriam.longo@unicampania.it (M.L.); 2Department of Advanced Medical and Surgical Sciences, University of Campania “Luigi Vanvitelli”, 80138 Naples, Italy; paolo.cirillo10@hotmail.com (P.C.); lorenzo828@virgilio.it (L.S.); katherine.esposito@unicampania.it (K.E.); 3Department of Precision Medicine, University of Campania “Luigi Vanvitelli”, 80138 Naples, Italy; mariateresa.vietri@unicampania.it; 4Department of Cardiothoracic and Respiratory Sciences, University of Campania “Luigi Vanvitelli”, 80138 Naples, Italy; antonio.bellastella@unicampania.it; 5Unit of Diabetes, University of Campania “Luigi Vanvitelli”, 80138 Naples, Italy

**Keywords:** autoimmune GHD, genetic GHD, anti-pituitary antibodies, lymphocytic hypophysitis, GH insensitivity

## Abstract

Growth hormone (GH), mostly through its peripheral mediator, the insulin-like growth factor 1(IGF1), in addition to carrying out its fundamental action to promote linear bone growth, plays an important role throughout life in the regulation of intermediate metabolism, trophism and function of various organs, especially the cardiovascular, muscular and skeletal systems. Therefore, if a prepubertal GH secretory deficiency (GHD) is responsible for short stature, then a deficiency in adulthood identifies a nosographic picture classified as adult GHD syndrome, which is characterized by heart, muscle, bone, metabolic and psychic abnormalities. A GHD may occur in patients with pituitary autoimmunity; moreover, GHD may also be one of the features of some genetic syndromes in association with other neurological, somatic and immune alterations. This review will discuss the impact of pituitary autoimmunity on GHD and the occurrence of GHD in the context of some genetic disorders. Moreover, we will discuss some genetic alterations that cause GH and IGF-1 insensitivity and the arguments in favor and against the influence of GH/IGF-1 on longevity and cancer in the light of the papers on these issues that so far appear in the literature.

## 1. Autoimmunity and Growth Hormone Deficiency (GHD) 

Growth hormone (GH), mostly through its peripheral mediator, the insulin-like growth factor 1(IGF1), plays its fundamental action in promoting linear bone growth in prepubertal age, but it continues playing an important role throughout life in the regulation of intermediate metabolism, trophism and the function of various organs, especially the cardiovascular, muscular and skeletal systems. Therefore, if a prepubertal GH secretory deficiency (GHD) is responsible for impaired growth and short stature, then a deficiency in adulthood identifies a nosographic picture classified as adult GHD syndrome, which is characterized by heart, muscle, bone, metabolic and psychic abnormalities [1]. Pituitary autoimmunity may play a pivotal role in favoring GHD both in children and in adults, as GH-secreting cells, alone or together with other pituitary hormone-secreting cells, may be aggressed by antipituitary antibodies (APA) in the context of lymphocytic hypophysitis (LYH) or of other forms of autoimmune hypophysitis [2,3,4,5,6,7,8] (Table 1A). Thus, to be able to search for APA in patients at risk may allow us to identify those prone to developing an autoimmune hypopituitarism [9]. Several methods have been proposed to detect these antibodies. Among these, indirect immunofluorescence is one of the most widely employed methods used to detect these antibodies in patients with suspected LYH; however, its sensitivity and specificity is low. Thus, the roles of antipituitary and antihypothalamus (AHA) antibodies are still discussed as methodological difficulties and also because several antigens have been suggested as the possible target of APA, but the true pituitary antigens are still a matter of discussion [10,11,12,13,14,15]. Consequently, in spite of the diffuse use of the immunofluorescence method, the results that appear in the literature so far are often conflicting, particularly due to the use of different human or animal substrates. This suggests caution against generalization of the results obtained with this method. We used cryostat sections of young baboon pituitary and hypothalamus glands to detect APA and AHA, due the difficulty of having human substrates in our disposal. However, we think that improvement of the specificity and sensitivity of this method may be obtained considering a predetermined cut-off of the titer and a particular kind of immunostaining, thus excluding the low titers and confounding immunostaining patterns. In our studies, we used as substrate cryostat sections of pituitary and hypothalamus glands from young baboon to detect APA and AHA by the simple indirect immunofluorescence method. In particular, fluorescein isothiocyanate conjugated with goat antihuman immunoglobulins was used to detect the presence of APA and AHA, considering a predetermined cut-off of the titer and the kind of immunofluorescence pattern used to improve the sensitivity and the specificity of the method. To this purpose, we considered APA positive at titers >1:8 and a particular immunostaining pattern, involving some but not all pituitary cells. This procedure allowed us to find out patients with autoimmune pituitary impairment and to foresee the kind of future hypopituitarism in those with pituitary function still normal, also suggesting that the occurrence of LYH is more frequent than that so far considered [2,3,4,5,6,7,8]. This procedure may allow more reliable results for diagnosing pituitary immunity, also by using animal substrates, especially when the results are validated by a second step with four-layer double immunofluorescence [8]. In fact, this method allows not only to detect APA and/or AHA, but also to identify the different cell lines targeted by these autoantibodies and to predict a possible specific hormonal deficiency. Using this method, the same pituitary section from young baboon is tested in a first step against the patient’s serum, and then against the fluorescein isothiocyanate (FITC) goat anti-human immunoglobulin sera; in the second step, it is tested against rabbit anti-sera, anti-GH, -ACTH, -TSH, -PRL, -FSH and -LH, separately, followed by rhodamine goat sera anti-rabbit IgG. The different color of anti-Ig conjugate against human sera and animal serum (green for FITC and red for rhodamine) allows for direct assessment of whether the patient’s serum and the animal’s sera stained the same or different pituitary cells. Using these methods, we were able to identify patients at risk of developing pituitary dysfunction [9] and to demonstrate that somatotrophs were the main target of APA in patients previously found to be APA-positive at a high titer [8]. Moreover, a longitudinal study in patients with autoimmune polyendocrine syndromes (APS) but without hypopitutarism at the enrolment, followed-up for 5 years, was able to identify those subsequently developing an autoimmune pituitary dysfunction [16], whereas, the characterization of pituitary cells targeted by antipituitary antibodies in patients with isolated autoimmune diseases without pituitary insufficiency helped us to foresee the kind of future hypopituitarism [17]. Concerning this, some patients with idiopathic GHD or with autoimmune endocrine diseases may show APA directed to GH-secreting cells [18], consequently favoring a GHD. These antibodies have been detected both in adults [17] and in children [18] with apparently idiopathic GHD. Moreover, APA have been detected at high frequency in a large cohort of Dutch patients with idiopathic isolated GHD or with idiopathic GHD among a multiple pituitary hormone deficiency, thus claiming the usefulness of searching for APA in patients with apparently idiopathic GHD to find out those with a silent form of autoimmune hypophysitis [19]. APA recognizing GH-secreting cells have been detected not only in children with idiopathic GHD but also in some children with idiopathic short stature (ISS) without GHD but whom subsequently develop this deficiency, suggesting that the detection of APA in children with ISS could identify those prone to subsequently developing an autoimmune GHD [18] (Table 1A). Figure 1 shows a sample positive for APA tested subsequently by a double four-layer immunofluorescence method in a study aimed at characterizing the pituitary-secreting cells targeted by these antibodies. 

However, not only pituitary autoimmunity, but also autoimmune processes involving the hypothalamus, may impair hormonal hypothalamic-pituitary secretions, including GH-Rreleasing Hormone GHRH ones. In fact, in some cases, AHA targeting several hypothalamic cells, including those GHRH-secreting ones, may cause an impairment of GHRH secretion and consequently a GHD [20]. Some diseases involving brain structures may be accompanied by a pituitary dysfunction, including GHD. An autoimmune GHD can occur in some patients following traumatic brain injury, in patients with Sheehan’s syndrome and in patients with acute infectious meningitis (Table 1A). 

The most frequent causes of traumatic brain injury (TBI) are road accidents (the main cause, which represents 50% of all cases) followed by falls, accidents related to violence, head injury linked to sport (hockey, football, soccer), combat sports (boxing and kickboxing) characterized by chronic repetitive head injuries and accidental war trauma, including explosion lesions [21,22,23,24,25,26,27,28,29,30,31,32,33,34,35]. Post-traumatic hypopituitarism is generally characterized by an isolated anterior pituitary hormone deficiency rather than multiple hormone deficiencies. Impairment of GH secretion and consequently of IGF-1 concentrations seems to be the most common early disorder after traumatic brain injury, both in the acute and chronic phase [25,29]. Concerning this, it has been suggested that impairment of GH and gonadotropin secretions is the most commonly pituitary disorder occurring post-TBI due to the more lateral location of somatotrophs and gonadotrophs at a pituitary level [25,26]. The pathophysiological basis of hypopituitarism secondary to traumatic brain injury is still discussed. The cause of the damage could be the hypoxic-ischemic insult, with subsequent oxidative stress and cytotoxicity leading to the death of neuronal cells by apoptosis or necrosis. Prospectively, in addition to the primary mechanical event, secondary insults (i.e., hypotension, hypoxia, hyperthermia and increased intracranial pressure due to skull fractures, edema and hemorrhage) and changes in cerebral flow and metabolism can contribute to hypothalamic-pituitary damage [26,27,28], which can contribute to perpetuating pituitary dysfunction. However, concerning the interrelationship between TBI and hypothalamic-pituitary autoimmunity, the first question to be satisfied is whether TBI may favor the development of the autoimmune process. It has been proposed that the release of sequestered pituitary or hypothalamic antigens from the necrotic hypothalamic-pituitary system after TBI may trigger an autoimmune response that is able to perpetuate neuroendocrine dysfunction, leading to late post-traumatic hypopituitarism [30,31,35]. Several studies reported the role of traumatic brain injury in triggering the neuroinflammatory and autoimmune process. In particular, some years ago Ankeny and Popovich underlined the potential mechanisms for CNS trauma-induced B cell activation and discussed the potential consequences of these injury-induced B cell responses. They concluded by hypothesizing that a subset of autoimmune B cell responses initiated by CNS injury could play a pathogenic role, thus suggesting that a targeted inhibition of B cell could improve recovery in cases of brain and spinal cord injury [36]. In the subsequent years, Zhang et al. reported their results in searching for the identification of serum autoantibody responses to brain-specific protein after TBI in humans. They found that TBI-evoked antibodies showed predominant immunoreactivity against a cluster of bands from 35–50 kDa on human brain immunoblots, which were identified as glial fibrillary acid protein (GFAP) and GFAP breakdown products. These antibodies showed an increase by 7 days after injury and were of IgG subtype predominantly. Changes in autoantibody levels were negatively correlated with outcome as measured by a Glasgow outcome scale- extended (GOS-E) score at 6 months, suggesting that TBI patients with greater anti-GFAP immune responses had worse outcomes [37]. The role of antibodies against GFAP in acute and chronic phases of TBI was subsequently further clarified by Wang et al., also opening the way to possible therapy [38]. However, not only negative effects have been attributed to TBI-evoked neuroinflammation and consequent autoimmunity. In fact, post-traumatic neuroinflammation may also promote brain recovery through the production of new neurons from neuronal stem/progenitor cells (NPCs) as demonstrated by experimental TBI in animals [38,39,40]. In fact, the neurogenic process is particularly stimulated by cytokines in some regions of the brain. In the hippocampus, for example, TBI robustly increases NPC proliferation, whereas injury-induced neuronal differentiation and survival of new neurons is far less pronounced [39,40]. Anyhow, a spontaneous cognitive recovery has been shown to be associated with granule neurons produced after TBI [41]. Moreover, a possible protective role had been also attributed to the TBI-evoked autoimmunity by previous studies [42,43]. In particular, a reduced neuronal loss favored by a neuroprotective T cell-dependent response evoked by CNS injury had been demonstrated in animals by Yoles et al. in 2001. They found that in transgenic mice overexpressing T cell receptors, ganglion cell survival after injury was higher [43]. With regards to the events leading to the post-TBI hypothalamic-pituitary deficiencies, considering all the findings previously discussed on this argument, it may be speculated that head trauma may trigger an ongoing cascade of vascular and histopathological alterations (necrotic, ischemic and hypoxic changes) also related to inflammation. Mediators of the inflammatory process may favor the activation of the immune system through the acceleration of neuronal cell necrosis, which allows us to unmask sequestered pituitary or hypothalamic antigens and the consequent production of respective autoantibodies that may contribute to late hypothalamic-pituitary dysfunction in TBI patients (35). These antibodies may precociously aggress GHRH and corticotropin-releasing hormone (CRH) neurons at the hypothalamic level and somatotrophs and gonadotrophs at the pituitary level with consequent impairment of the respective hormone secretions.

A similar cascade of events evoking an autoimmune aggression to the hypothalamus–pituitary axis impairing GH secretion may occur in patients with Sheehan’s syndrome (SS). In fact, the presence of APA and AHA has been detected in some of these patients [44] even many years after the onset of hypopituitarism. In particular, AHA and APA were detected at a high titer in 8 and in 7, respectively, of 20 patients with SS. However, none of these patients had AHA immunostaining vasopressin-secreting cells, but only releasing factor-secreting cells. This suggests that in women with this syndrome, an autoimmune process involving both the hypothalamus and the pituitary gland, triggered by cellular damage caused by ischemia, may contribute to late pituitary dysfunction involving GH secretion directly or through the inhibition of GHRH secretion [44]. Pituitary dysfunction may occur also after acute bacterial or viral meningitis. A study by Tanrivedi et al. investigated whether autoimmune mechanisms could play a role in the pathogenesis of acute meningitis-induced hypopituitarism, searching prospectively for APA and AHA in 16 affected patients in the acute phase and at 6 and 12 months after the acute meningitis. A GHD was diagnosed in 18.7% of patients in acute phase. At 12 months, 6 patients had GHD that was isolated or associated with other pituitary hormone deficiencies. The occurrence of AHA and APA positivity was substantially high in these patients, ranging from 35 to 50%. This seems to suppose a possible role of autoimmunity in the pathogenesis of pituitary dysfunction after acute infectious meningitis [45]. Taking this into account, we think that searching for AHA and APA in some conditions may help to avoid an underestimation of autoimmune causes of pituitary dysfunction and, in particular, of GHD. Thus, we suggest to search for APA and AHA in patients with apparently idiopathic GHD, isolated or associated with other pituitary hormone deficiencies, especially if belonging to an autoimmune polyglandular syndrome (APS), in patients with post-traumatic hypopituitarism or with some brain diseases involving hypothalamic-pituitary axis. This could favor an early diagnosis of hypothalamic-pituitary autoimmunity, allowing, if possible, to interrupt with appropriate therapy the progression to a clinically overt GHD. In particular, in patients with short stature positive for APA and with GHD in childhood, submitted to replacement GH therapy, it is advisable to retest pituitary function and APA detection after the stopping of therapy in the transition age. In fact, some patients positive for APA at middle but not at high titer may show a disappearance of APA and a remission of GHD after GH replacement therapy. Moreover, as APA may shift their pituitary target in the transition age from somatotrophs to gonadotrophs, an early characterization of APA by double immunofluorescence is advisable in GHD patients positive for APA showing delayed puberty to allow an early diagnosis and an appropriate therapy, thus preventing the progression from a delayed puberty to clinically overt hypogonadotropic hypogonadism [46]. Moreover, to search for APA and AHA may contribute to diagnosing the different form of autoimmune hypophysitis in patients at risk also in potential or subclinical age. In addition to the classic forms of LYH (lymphocytic adenohypophysitis, lymphocytic infundibulo-neurohypophysitis, lymphocytic panhypophysitis), some new forms of pituitary autoimmune disease that may cause also GHD have recently been described, namely the IgG4-related hypophysitis, the anti-CTL-4 hypophysitis and the anti-PIT-1 hypophysitis [5,6,47,48,49] (Table 1A). The IgG4-related hypophysitis is characterized by a massive infiltration of the pituitary and stalk by IgG4-secreting plasma cells associated with a variable degree of fibrosis and high plasma IgG values. This hypophysitis is included together with several other diseases in the spectrum of IgG4-related diseases, an increasingly recognized syndrome of unknown etiology, which includes a collection of disorders that share specific histopathological, serological and clinical characteristics [47]. A further form of secondary autoimmune hypophysitis has recently been described in patients undergoing immunotherapy with antibodies to CTLA4 (cytotoxic T-lymphocytic-associated antigen-4) for melanoma or other malignancies [48]. In such cases, the hypophysitis represents one of the immune-related adverse events (IRAEs). The anti-PIT-1 hypophysitis (anti-PIT-1 antibody syndrome) is characterized by acquired and specific GH, prolactin and TSH deficiencies. This disorder is associated with a thymoma or other neoplasm that ectopically expresses PIT-1 protein. In this case, the circulating anti-PIT-1 antibody is a disease marker, and PIT-1-reactive cytotoxic T cells (CTLs) play a pivotal role in disease development. Although several underlying mechanisms for pituitary autoimmunity have been proposed and previously extensively discussed, these findings suggest a new possible pathophysiological aspect highlighting the importance of paraneoplastic syndrome as a cause of pituitary autoimmune diseases [49].

## 2. GHD in the Context of Genetic Disorders

GHD may also be one of the features of some genetic syndromes associated sometimes with other neurological, somatic and immune alterations, causing, if misdiagnosed, further distress in affected patients [50,51] (Table 1B). Mutations in the GH1 and GHRH genes shed light on the phenotype and pathogenesis of isolated GHD, whereas mutations in transcription factors such as HESX1, PROP1, POU1F1, LHX3, LHX4, GLI2 and SOX3 are involved in combined pituitary hormone deficiencies. Depending upon the expression patterns of these molecules, the phenotype may consist of isolated pituitary dysfunction or more complex disorders such as septo-optic dysplasia and holoprosencephaly [49,50,51,52]. A link between genetic disorders and autoimmunity has been described in Prader–Willi syndrome (PWS), a genetic condition caused by loss of the paternal copy of a region of imprinted genes on chromosome 15, in which developmental delay is associated with severe muscular hypoplasia and hyperphagia leading to severe obesity. The phenotype is most probably due to hypothalamic dysfunction, which is also responsible for GH, TSH and ACTH deficiencies and central hypogonadism [53]. A study investigating the role of autoimmune pituitary involvement in 55 adults with this syndrome discovered that about 30% of them were positive for APA. The authors concluded that, although the presence of these antibodies could only be an “epiphenomenon”, their results suggested that autoimmune mechanisms might contribute, at least in part, to the pituitary dysfunction of Prader–Willi syndrome, and they claim to search for APA in these patients to clarify the role of pituitary autoimmunity in their pituitary dysfunction [54]. GH treatment in affected patients has been demonstrated to improve muscle bulk, reduce fat mass and increase levels of physical and cognitive activities other than to improve body growth [55].

The hyper-immunoglobulin M syndromes (HIGM) are a heterogeneous group of genetic disorders, due to several mutations of activation-induced cytidine deaminase (AICDA) inducing extremely elevated IgM and significantly decreased IgG and IgA, which has been rarely reported to be associated with GHD [56]. However, a GHD associated with other physical alterations has been described in a nine-year-old girl with short stature and with a new AICDA mutation. She responded well to systemic corticosteroid and to Ig and GH replacement therapy. The authors concluded that the mutation analysis could clarify the pathophysiological aspects of this syndrome and improve the diagnosis of HIGM patients, addressing the most appropriate therapy to the affected patients [57]. The region p13 of the chromosome 17 is a region of genomic instability that is linked to different rare neuro-developmental genetic diseases, depending on whether a deletion or duplication of the region has occurred [58,59]. The 17p13.1 syndrome is a rare genetic disorder characterized by short stature and GHD associated with intellectual disability, facial dysmorphisms and obesity. Among the alterations of the neuroendocrine structures, a possible pituitary dysfunction causing GHD may occur in patients with microdeletion of 17p13.1. Recently, Leka-Emiri and coworkers described a case of a child with a maternally inherited 17p31.1 microdeletion presenting with apparently familial short stature but with low IGF1 for their age and impaired GH response to appropriate stimuli. They concluded affirming that, although familial short stature is considered a normal variation of growth retardation, hormonal and genetic investigation is essential in the etiological diagnosis, allowing for an appropriate GH replacement therapy in those with GHD [59]. A heterozygous deletion at chromosome 17p11.2 region that includes RAI1 (or a heterozygous intragenic RAI1 pathogenic variant) characterizes the Smith-Magenis syndrome [60]. This syndrome is characterized by distinctive physical features (particular facial features that progress with age), developmental delay, cognitive impairment, behavioral abnormalities, sleep disturbance and childhood-onset abdominal obesity. The behavioral phenotype, including significant sleep disturbance, stereotypies and maladaptive and self-injurious behaviors, is generally not recognized until age 18 months or older and continues to change until adulthood. Hearing loss and skeletal, ophthalmologic, cardiac and renal anomalies are usually present, whereas, among the endocrine alterations, hypothyroidism and GHD frequently occur, causing more severe mental and somatic impairment if not corrected early with appropriate replacement therapy [60]. Instead, microdeletions and microduplications concerning the 17p13.3 region can result in either isolated lissencephaly sequence (ILS) or Miller-Dieker syndrome (MDS) [58,59]. Both conditions are associated with a smooth cerebral cortex, or lissencephaly, which leads to developmental delay, intellectual disability and seizures. However, patients with MDS have larger deletions than patients with ILS, resulting in additional symptoms such as poor muscle tone, congenital anomalies, abnormal spasticity and craniofacial dysmorphisms. In both conditions, developmental delay may be related to GHD among a complex picture of encephalic and somatic alterations. A possible GHD can occur in patients with cystic fibrosis [61,62]. This is an incurable, chronic disease that cause severe damage to respiratory and digestive tracts and is the most common genetically inherited disease among Caucasian population. This disease is caused by a defect in cystic fibrosis (CF) genes, the so-called mutations in cystic fibrosis transmembrane conductance regulator (CFTR) gene population. At present over 100,000 people suffer from this disease [61]. Patients with CF may present with signs and symptoms that overlap with those of adult GHD syndrome. A recent study investigating the hormonal pattern of 50 clinically stable adult CF patients showed an impaired GH response to GHRH+Arginine in 16 of them (32%). GHD was severe in 7 and partial in 9 patients. The authors concluded that adult patients with CF may show GHD [62]. Also, in these cases, an appropriate GH replacement therapy is advisable to avoid heart, muscle, bone, metabolic and psychic abnormalities, all pictures of adult GHD syndrome [1]. Also, for girls with Turner syndrome, a 45/X syndrome characterized by growth failure, gonadal insufficiency and somatic and internal organ alterations, may have some benefit from GH therapy, even if not belonging to the classical GHD (Table 1B). In fact, in these cases, the doses used are usually higher than those used in children with GHD and the results are still discussed. A recent paper investigated the benefits and adverse effects of GH treatment in women with Turner syndrome (TS), comparing a group of 33 TS patients treated in childhood with GH to a group of 124 TS patients who did not receive GH. Treated patients were significantly taller and had a better lipid profile and lower prevalence of arterial hypertension than untreated patients. However, they also had low thrombocyte counts, a greater prevalence of retrognatism, nail anomalies and elevated creatinine levels, especially when the GH treatment was delayed or prolonged. They concluded that GH treatment in children with TS has benefits in adulthood but adverse effects may occur, especially in girls with treatment that is delayed or is too long [63].

## 3. Genetic Alteration Affecting GH and IGF-1 Actions

The genetic defects affecting GH and/or IGF-1 actions resulting in short stature can be classified in five categories: (a) GH insensitivity by defects affecting the GH receptor; (b) alteration of the intracellular GH signaling pathway (STAT5B, STAT3, IKBKB, IL2RG, PIK3R1); (c) altered synthesis of IGFs; (d) altered transport/bioavailability of IGFs (IGFALS,PAPPA2); (e) IGF-1 insensitivity by defects affecting the IGF1 sensitivity linked to mutation of IGF1 receptor [64] (Table 1C). All these categories show GH/IGF-1 deficiency or insensitivity. Zvi Laron first described a syndrome with complete GH insensitivity in patients with characteristic features of GHD, but presenting increased levels of GH [65]. These patients are all characterized by typical appearance such as dwarfism, facial phenotype, obesity and hypogenitalism. Moreover, in affected patients, GH insensitivity, caused by deletion or mutations of the GH receptor gene, may be associated with several clinical signs of immune dysfunction and autoimmune dysregulation, indicating a possible interrelationship between the GH-IGF-1 system and the mechanisms evoking immune dysregulation. This occurs especially in patients with molecular defects in the intracellular GH signaling pathway (STAT5B, STAT3, IKBKB, IL2RG, PIK3R1) [64]. Interestingly, patients with Laron syndrome do not develop cancer [66,67,68,69,70] and recent studies on genome-wide profiling of patients with this syndrome has been able to identify novel cancer protection pathways, opening the way to new developments in oncology [69,70]. These results, together with those of the studies in centenarians [70,71], seem to indicate a preventing effect of reduced secretion or lacking action of GH-IGF-1 on some diseases with increased life expectance. However, a recent study published in September 2019 suggests caution against generalization of these assumptions. The authors used a protocol intended to regenerate the thymus in healthy aging men by GH administration. They observed protective immunological changes, improved risk indices for many age-related diseases and a mean epigenetic age approximately 1.5 years less than baseline after 1 year of treatment. The GrimAge predictor of human morbidity and mortality in these patients showed a 2-year decrease in epigenetic vs. chronobiological age that persisted six months after discontinuing treatment [72]. Thus, this seems to suggest caution in the use of possible interventions aimed at down-regulating activity of the GH-IGF-1/insulin pathway for the extension of human life span. In conclusion, it can be affirmed that replacement GH therapy in GHD short children and in adults with GHD has been shown to be safe when no other risk factors for malignancy are present. Nevertheless, the use of GH in cancer survivors and in short children with RASopathies, (caused by germline pathogenic variants in genes that encode RAS pathway proteins, which make affected patients at increased risk of cancer [73]), chromosomal breakage syndromes or DNA-repair disorders should be carefully evaluated owing to an increased risk of recurrence of primary cancer, or second neoplasm in these individuals [72].

## Figures and Tables

**Figure 1 ijms-21-01392-f001:**
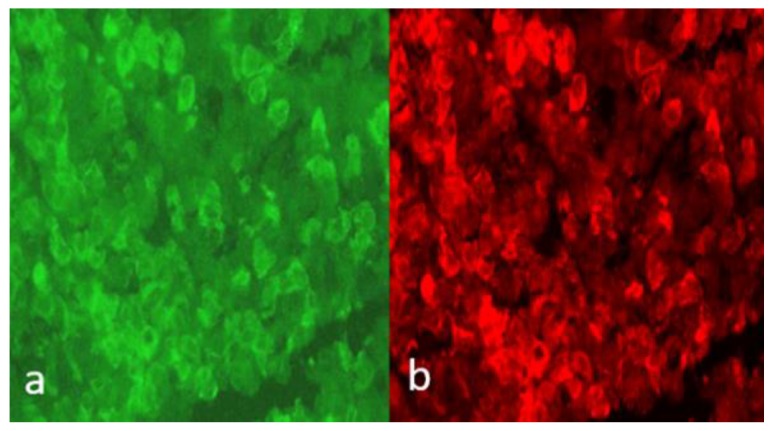
Immunofluorescence in cryostat sections of young baboon anterior pituitary gland tested against the serum of a patient with isolated growth hormone deficiency GHD, in a primary step adding FITC goat sera anti-human immunoglobulins (left panel: (**a**) = simple immunofluorescence) and in a second immunostaining step adding rabbit antisera anti-GH followed by rhodamine goat sera anti rabbit IgG (right panel: (**b**) = double four-layer immunofluorescence). The overlapping color in the same cells, green in left panel and red in right panel, indicates that the cells immunostained by APA are the somatotrophs [18]. Scale bars = 40×.

**Table 1 ijms-21-01392-t001:** Growth hormone and IGF-1 deficiency or insensitivity in autoimmune and genetic diseases.

A: GHD and Autoimmunity	B: GHD and Genetic Disorders	C: GH and IGF-1 Insensitivity and Genetic Disorders
**Lymphocytic hypophysitis:** (a) Lymphocytic adenohypophysitis (b) Lymphocytic infundibulo-neurohypophysitis (c) Lymphocytic panhypophysitis (d) IgG4-related hypophysitis (e) Anti-CTLA-4 hypophysitis (f) Anti-PIT-1 hypophysitis **Pituitary autoimmunity linked to TBI** **Pituitary autoimmunity in Sheehan’s syndrome** **Pituitary autoimmunity after acute meningitis**	Prader Willi syndrome Hyper-immunoglobulin M syndrome 17p13.1 syndrome Smith-Magenis syndrome Isolated Lissencephaly Sequence Miller-Dieker syndrome Cystic fibrosis Turner syndrome	Defects of the GH receptor Altered intracellular GH signaling pathway Altered synthesis of IGFs Altered transport/bioavailability of IGFs IGF-1 insensitivity linked to mutation of IGF1 receptor Laron syndrome

GHD: Growth hormone deficiency; TBI: traumatic brain injury; IGF-1: Insulin like growth factor-1.

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
