# Peer review of "Impact of Pituitary Autoimmunity and Genetic Disorders on Growth Hormone Deficiency in Children and Adults"

_ijms, 2020, doi:10.3390/ijms21041392_

Round 1

Reviewer 1 Report

Growth hormone deficiency (GHD) is a medical condition due to not enough growth hormone (GH). Some cases are associated with a lack of other pituitary hormones, in which case it is known as combined pituitary hormone deficiency. A prepubertal GHD is responsible for short stature, a deficiency in adulthood identifies as adult GHD syndrome, characterized by heart, muscle, bone, metabolic and psychic abnormalities. GH effects are mediated by insulin-like growth factor 1 (IGF-1). The most common treatment for growth hormone deficiency in both children and adults is growth hormone therapy — injections of growth hormone into the body. GH plays a very important role in maintaining the homogeneity of tissues during normal development or after injury. A separate problem is growth hormone therapy when there is no deficiency (e.g. in case of GH and IGF-1 insensitivity). It is widely acknowledged that in acromegaly, excessive GH causes hypertensive and diabetogenic effects. GH replacement therapy also induces insulin resistance. The growth hormone might be a prohormone rather than a hormone since it is proteolytically cleaved in a tissue-specific manner, giving rise to shorter GH forms whose activities are still unknown. As a pleiotropic hormone, GH has a number of endo-, auto-, and paracrine effects on practically every tissue. The effects of childhood growth hormone treatment are not always beneficial to the patient, in adulthood. The authors of the work entitled: “IMPACT OF PITUITARY AUTOIMMUNITY AND GENETIC DISORDERS ON GROWTH DEFICIENCY IN CHILDREN AND ADULTS” Giuseppe Bellastella, Maria Ida Maiorino, Miriam Longo, Paolo Cirillo, Lorenzo Scappaticcio, Maria Teresa Vietri, Antonio Bellastella, Katherine Esposito, Annamaria De Bellis reviewed the literature and systematized the occurrence of the above syndromes Growth hormone and IGF-1 deficiency as a consequence of the autoimmune inflammatory process of the pituitary gland, Growth hormone and IGF-1 insensitivity in autoimmune and genetic diseases as well as GH / IGF-1 insufficient sensitivity (in receptor resistance) for various genetic disorders. The subject of the work is very interesting and current. The division presented by the authors is logical and coherent in both aspects: methodological and didactic. Despite encouraging results, diagnostics and early recognition of processes leading to the diagnosis of individual units are currently not sufficiently developed. In my opinion, extending the thesis on the issues of the arguments in favor and against the influence of GH / IGF-1 on longevity and cancer in the light of the papers on these issues so far appeared in the literature, it is not necessary. It can be the subject of a separate work. The title of the work seems incomplete. The word 'hormone' or 'hormone secretion' should be included (Line 3).

Author Response

We agree with this referee about the lack of “hormone” in the title. It has been added in the revised paper.  Regarding the arguments in favour and against the influence of GH/IGF1 on longevity and cancer, also respecting the referee’s opinion, we think these arguments may be of some interest for this review.

Reviewer 2 Report

Manuscript ID: ijms-685718

IMPACT OF PITUITARY AUTOIMMUNITY AND GENETIC DISORDERS ON GROWTH
DEFICIENCY IN CHILDREN AND ADULTS

The review describes multiple manifestations of growth hormone deficiency. On average all the data are interesting and useful for physicians and researchers.

Minor remarks:

The figure is not very convincing, better to remove it at all as well as to remove the description of the method (In our studies, we used as substrate cryostat sections of pituitary and hypothalamus glands from young baboon to detect APA and AHA by simple indirect immunofluorescence method. In particular, fluorescein isothiocyanate conjugated with goat antihuman immunoglobulins was used to detect the presence of APA and AHA, considering a predetermined cut-off of the titre and the kind of immunofluorescence pattern to improve the sensitivity and the specificity of the method. To this purpose, we considered APA positive at titres > 1:8 and a particular immunostaining pattern, involving some but not all pituitary cells. This procedure allowed us to find out patients with autoimmune pituitary impairment and to foresee the kind of future hypopituitarism in those with pituitary function still normal, also suggesting that the occurrence of LYH is more frequent than that so far considered[2-8]. This  procedure may allow more reliable results for diagnosing pituitary immunity, also by using animal substrates, especially when the results are validated by a second step with four-layer double immunofluorescence [8]. In fact, this method allows not only to detect APA and/or AHA but also to identify the different cell lines targeted by these autoantibodies and predict a possible specific hormonal deficiency. Using this method, the same pituitary section from young baboon is tested in a first step against the patient’s serum and then fluorescein isothiocyanate (FITC) goat anti-human immunoglobulin sera; in the second step, against rabbit anti-sera, anti-GH, -ACTH, -TSH, -PRL, -FSH and –LH, separately followed by rodamine goat sera anti-rabbit IgG. The different color of anti-Ig conjugate against human sera and animal serum (green for FITC and red for rodamine),allows for direct assessment of whether the patient’s serum and the animal’s sera stained the same or different pituitary cells). It is enough to keep a citation.

Also it may be good to add sub-headings, the text id slightly difficult to read.

Author Response

An accurate description of methods used in our lab and a figure were added to our initial version following  the previous suggestion  to make the paper more complete. In particular, we think useful for the reader a figure showing the identification by a double immunofluorescence of the hormone-secreting cells targeted by APA

We agree with this referee on difficulty to read the title. Actually, it was lacking of the word “hormone” after the word growth. It has been added in the revised version.

Reviewer 3 Report

This review describes the role of pituitary autoimmunity on the GH deficiency (GHD) and the occurrence of GHD in the context of several genetic disorders. In addition, the effect of GH/IGF-1 system on the modulation of longevity is discussed.

This is an interesting paper written by internationally recognized experts in the field of pituitary autoimmunity.

Few minor comments are suggested to improve on the submitted manuscript:

Patients with Laron syndrome do not develop cancer and have a high insulin sensitivity. In addition also centenarians are characterized by extremely high HOMA2-S (please see PMID: 22983440). This aspect should be discussed. Table 1 should be better structured in order to help reader understand. Line 336 “…short children with RSApathies…”. Please define RSApathies. A short conclusion should be included at the end of the manuscript.

Author Response

Table 1 has been better structured as suggested.

We do not think to further discuss the argument on the centenarians also because the Reviewer 1 suggest that “extending the thesis on the issues of the arguments in favor and against the influence of GH / IGF-1 on longevity and cancer in the light of the papers on these issues so far appeared in the literature, it is not necessary”

Actually, the correct form is RASopathies( RSApathies was an error of typewriting). It has been corrected in the revised paper, also defining these syndromes and adding an appropriate reference(73) in bibliography.